# Team-family conflicts over end-of-life decisions in ICU: A survey of French physicians' beliefs

Mikhael Giabicani[1,2]*, Laure Arditty[3], Marie-France Mamzer[2,4], Isabelle Fournel[5], Fiona Ecarnot[6,7], Nicolas Meunier-Beillard[5,8], Fabrice Bruneel[9], Emmanuel Weiss[1], Marta Spranzi[10], Jean-Philippe Rigaud[11,12], Jean-Pierre Quenot[5,13,14,15]

1 Department of Anaesthesiology and Critical Care, Beaujon Hospital, DMU Parabol, AP-HP Nord, and Université Paris Cité, Paris, France, 2 Centre de Recherche des Cordeliers, Sorbonne Université, Université Paris Cité, Inserm, Laboratoire ETREs, Paris, France, 3 Service de Réanimation, Centre Hospitalier Intercommunal des Alpes du Sud, Gap, France, 4 Unité Fonctionnelle d'Ethique Médicale, Hôpital Necker-Enfants Malades, AP-HP, Paris, France, 5 CHU Dijon Bourgogne, INSERM, Université de Bourgogne, CIC 1432, Module Épidémiologie Clinique, Dijon, France, 6 Department of Cardiology, University Hospital Besançon, Besançon, France, 7 EA3920, Université de Bourgogne-Franche Comté, Besançon, France, 8 DRCI, USMR, CHU Dijon Bourgogne, Dijon, France, 9 Intensive Care Unit, Versailles Hospital Center, Le Chesnay, France, 10 Center for Clinical Ethics, AP-HP, Paris and Université de Versailles Saint-Quentin en Yvelines, Versailles, France, 11 Service de Médecine Intensive Réanimation, CH de Dieppe, Dieppe, France, 12 Espace de Réflexion Éthique de Normandie, CHU de Caen, Caen, France, 13 Service de Médecine Intensive-Réanimation, CHU Dijon-Bourgogne, Dijon, France, 14 Equipe Lipness, Centre de Recherche INSERM UMR1231 et LabEx LipSTIC, Université de Bourgogne-Franche Comté, Dijon, France, 15 Espace de Réflexion Éthique Bourgogne Franche-Comté (EREBFC), Dijon, France

* mikhael.giabicani@aphp.fr

**Data Availability Statement:** All relevant data are within the paper and its Supporting Information files.

## Abstract

### Introduction

Conflicts between relatives and physicians may arise when decisions are being made about limiting life-sustaining therapies (LST). The aim of this study was to describe the motives for, and management of team-family conflicts surrounding LST limitation decisions in French adult ICUs.

### Methods

Between June and October 2021, French ICU physicians were invited to answer a questionnaire. The development of the questionnaire followed a validated methodology with the collaboration of consultants in clinical ethics, a sociologist, a statistician and ICU clinicians.

### Results

Among 186 physicians contacted, 160 (86%) answered all the questions. Conflicts over LST limitation decisions were mainly related to requests by relatives to continue treatments considered to be unreasonably obstinate by ICU physicians. The absence of advance directives, a lack of communication, a multitude of relatives, and religious or cultural issues were frequently mentioned as factors contributing to conflicts. Iterative interviews with relatives and proposal of psychological support were the most widely used tools in attempting to

**Funding:** The author(s) received no specific funding for this work.

**Competing interests:** The authors have declared that no competing interests exist.

**Abbreviations:** ADs, Advance directives; EOL, End-of-life; ICU, Intensive care unit; LST, Life-sustaining therapies; RESC, Réseau de recherche en éthique en soins critiques; SD, Standard deviation.

resolve conflict, while the intervention of a palliative care team, a local ethics resource or the hospital mediator were rarely solicited. In most cases, the decision was suspended at least temporarily. Possible consequences include stress and psychological exhaustion among caregivers. Improving communication and anticipation by knowing the patient's wishes would help avoid these conflicts.

## Conclusion

Team-family conflicts during LST limitation decisions are mainly related to requests from relatives to continue treatments deemed unreasonable by physicians. Reflection on the role of relatives in the decision-making process seems essential for the future.

## Introduction

Although knowledge in the field of intensive care has considerably progressed in recent decades, mortality among patients hospitalized in the intensive care unit (ICU) remains approximately 20% [1–3]. It is estimated that LST limitation decisions are made in at least half of all deaths that occur in intensive care [4, 5].

In France, as in numerous other countries [6], LST limitation decisions are made within a strict legislative framework that involves a collegial deliberative process [7–9], which must take into account the patient's wishes, in particular through written advance directives (ADs) (see S1 File for details of French legislation). However, situations where ADs are known, and/or where the patient is able to express himself/herself are extremely rare [3], and thus, the relatives are often the only persons who can relate the patient's wishes. Under French law, relatives have no decision-making role and their opinion is only advisory [7, 8]. The final decision is made by the physician (or medical team), who bears the responsibility for the decision, and guarantees its application.

One of the fundamental principles underlying decisions to withhold or withdraw treatment in ICU is the refusal of "unreasonable therapeutic obstinacy" [10–12]. The evaluation of unreasonable therapeutic obstinacy requires an assessment of each situation that is partly based on subjective, or even emotional elements and often different between the patient, the relatives, and the medical team [13]. In some cases, these different visions can give rise to disagreements or conflicts surrounding LST limitation decisions between relatives and caregivers. While a worldwide professional consensus has been developed regarding the major ethical end-of-life (EOL) principles, marked variations exist globally, as well as differences within each country and society [6, 12]. In France in particular, the law was originally intended to ensure that patients would not be subjected to unreasonable obstinacy [7, 8, 14, 15]. The legislation subsequently allowed for the possibility for relatives to oppose a medical decision to limit LST by appealing to a judge, suggesting a paradigm shift [16]. Recent case reports of legal proceedings in the literature clearly highlight these problematic situations [17, 18].

Some data on conflict exist in the literature and have led to the development of frameworks and recommendations, as in the United States and Canada [19, 20]. To the best of our knowledge, the current French data on conflicts in the ICU address the issue in a general way, and do not specifically focus on LST limitation decisions [21]. It remains unclear what the current causes of conflicts around LSTs are in the ICU setting, and how these conflicts are managed. Before considering possible practice recommendations, it seems important to document these conflicts in the French medical and legislative context.

The main objective of this study was to describe the sources and management of team-family conflicts surrounding LST limitation decisions in French ICUs. Secondary objectives were to describe physicians' beliefs about the facilitating factors and potential consequences of these conflicts.

## Methods

### Study design and population

We performed a national, prospective, observational, multicenter survey of practices among French ICU physicians using an electronic questionnaire sent between June 22, 2021 and October 4, 2021. This survey was performed through a Research Network in Ethics in Critical Care ("Réseau de recherche en éthique en soins critiques", RESC). The RESC is a network for disseminating information and calling for participation in studies on the topic of ethics in critical care so as to ensure representativeness in terms of type and size of the participating ICUs, and in terms of practices of intensive care physicians.

University and non-university intensive care physicians referenced within the RESC network were contacted electronically to complete the questionnaire. As points of view may vary from one physician to another, several physicians from the same ICU could answer the questionnaire. Only one response to the questionnaire per physician was accepted.

### Study questionnaire

A questionnaire was developed comprising 20 questions about respondents' beliefs and practices in terms of sources and management of conflicts.

The questionnaire was developed by two intensive care physicians. The acquisition of the empirical data underpinning the questionnaire items followed an exploratory phase with a panel of intensive care physicians, two consultants in clinical ethics (a doctor of philosophy and a physician, Clinical Ethics Center for Paris University Hospitals (AP-HP), France) and a professor of medical ethics (Paris Cité University). This exploratory phase was conducted as a qualitative study using *in situ* observations and semi-structured interviews (open-ended questions in one-to-one interviews) to determine, in combination with previous qualitative data from the literature [22], the potentially important elements for physicians regarding situations of conflict surrounding LST limitation.

The questionnaire and the possible answers to each question were then modified and enriched during a focus group comprising intensive care physicians working in academic and/ or non-academic hospitals, a sociologist and a statistician.

Finally, the questionnaire was tested on a new panel of 13 intensive care physicians to judge the understanding and relevance of each item of the questionnaire, as well as the reproducibility of the answers obtained after several proofreadings (test/retest). Some items were rephrased to achieve maximum readability before the final validation of the questionnaire.

The final survey consisted of 20 questions divided into 4 main themes: origin and manifestations of the conflict; conflict management; impact of the conflict; potential ways to prevent conflicts. Among the questions, 7 were on a scale of frequency (yes, all of the time; yes most of the time; sometimes; rarely; never), 4 were scored using a 5-level Likert scale ranging from "completely agree" (+2) to "completely disagree" (-2) and 9 were single or multiple choice questions. Finally, we also recorded the main demographic characteristics of physicians (age, sex, number of years' experience as an intensive care physician). With the exception of the demographic characteristics, the responses to the questionnaire were exclusive.

The questionnaire is provided in S1 Table.

## Distribution of the questionnaire and data collection

The anonymized questionnaire was distributed via the LimeSurvey platform. The distribution of the survey and the data management were performed by the Clinical Investigation Center of University hospital of Dijon (certified ISO 9001128 V2015).

## Statistical analysis

Qualitative variables are expressed as numbers (percentages) and were compared using the Chi square or Fisher's exact test as appropriate. It should be noted that the response categories "yes, all the time" and "yes, most of the time" were merged, as were the categories "rarely" and "never". For responses on Likert scales, we considered a response rate to be relevant when it was above 50%. For continuous measurements, data are presented as mean ± standard deviation (SD).

Associations between physician grade, junior (≤2 years of critical care experience) or senior (>2 years of critical care experience), and conflict management were explored by univariate analysis.

A p-value <0.05 was considered statistically significant. All analyses were performed using SPSS version 16.0 (SPSS Inc., Chicago, IL).

**Table 1. Characteristics of the participating intensive care units and physicians.**

| Participants characteristics (number of respondents) | n (%) |
|---|---|
| **Age** (n = 148) | |
| • ≤34 years | 35 (24) |
| • 35–49 years | 75 (51) |
| • ≥50 years | 38 (26) |
| **Male sex** (n = 158) | 107 (68) |
| **Grade of respondent** (n = 159) | |
| • Junior physician (≤2 years) | 19 (12) |
| • Senior physician (>2 years) | 140 (88) |
| **Number of years of ICU practice** (n = 140) | |
| • ≤4 years | 21 (15) |
| • 5–9 years | 42 (30) |
| • ≥10 years | 77 (55) |
| **Type of hospital** (n = 160) | |
| • Non-academic | 74 (46) |
| • Academic | 70 (44) |
| • Private | 4 (3) |
| • Other | 12 (7) |
| **Type of ICU** (n = 156) | |
| • Mixed | 91 (58) |
| • Medical | 43 (28) |
| • Surgical | 14 (9) |
| • Pediatric | 6 (4) |
| • Other | 2 (1) |

ICU, Intensive Care Unit. Data are expressed as number (percentage).

### Ethics statement

The ethics committee of the French Society of Anaesthesia, Critical Care and Perioperative Medecine approved this study (IRB 00010254-2022-014) and waived the need for consent.

## Results

### Study population

Among the 186 intensive care physicians in the RESC network, 160 (86%) physicians from 85 ICUs answered all the questions. The characteristics of the responding physicians are displayed in Table 1. They were mostly men (sex ratio 1:2), aged 35 to 49 years, with more than 10 years' experience in ICU practice. They mainly exercised in mixed or medical ICUs, in academic or non-academic public hospitals.

### Origin and manifestations of the conflict

**Motives for the conflict.** The main reason for conflicts about LST limitation decisions was related to relatives' opposition to the decision, with relatives believing, unlike the physicians, that the patient is not in a situation of unreasonable obstinacy (66% of respondents). A small minority (5%) of physicians reported conflicts linked to the caregiving team's refusal to consider an LST limitation procedure.

The relatives objected to a decision to withdraw or withhold treatment in 64% and 36% of cases respectively. The results regarding motives of the conflict are summarized in Fig 1.

**Manifestation of the conflict.** Respondents stated that conflicts arise mainly during discussions between relatives and the medical (85%) or paramedical (71%) team; before (37%) or after the collegial meeting (57%). Sixty-two percent of physicians reported aggressiveness or even physical or verbal threats towards caregivers.

### Conflict management

The elements of conflict management are displayed in Fig 2. Iterative interviews with relatives are the most widely used and useful tool in trying to resolve the conflict. Offering psychological support and proposing to call on a physician from outside the department are also widely used techniques. Conversely, the intervention of a mobile palliative care team, a local ethics resource or the hospital mediator are rarely used.

In the vast majority of cases, the decision is not applied as usual when there is team-family conflict surrounding the decision. Only 19% of physicians reported that they would apply the decision without taking the conflict into account. For more than 85% of respondents, the decision is most often reassessed during new collegial meetings or applied gradually, and sometimes even suspended.

Sixty-six percent of physicians declared that the legal decision-making process is more scrupulously followed when a conflict exists.

Finally, for 66% of respondents, the conflict most often subsides before the patient's death or discharge. Despite the conflict, 18% of physicians believe that the death of the patient after the LST limitation ultimately represents a form of relief for the relatives.

### Potential ways to prevent conflict

Among the suggestions for preventing conflicts, four main elements were highlighted by the physicians: conducting family interviews in a formal way, in a dedicated room, with dedicated time; systematically searching for ADs on ICU admission; setting up free and unlimited visiting hours to facilitate the presence of relatives with the patient; providing families with an

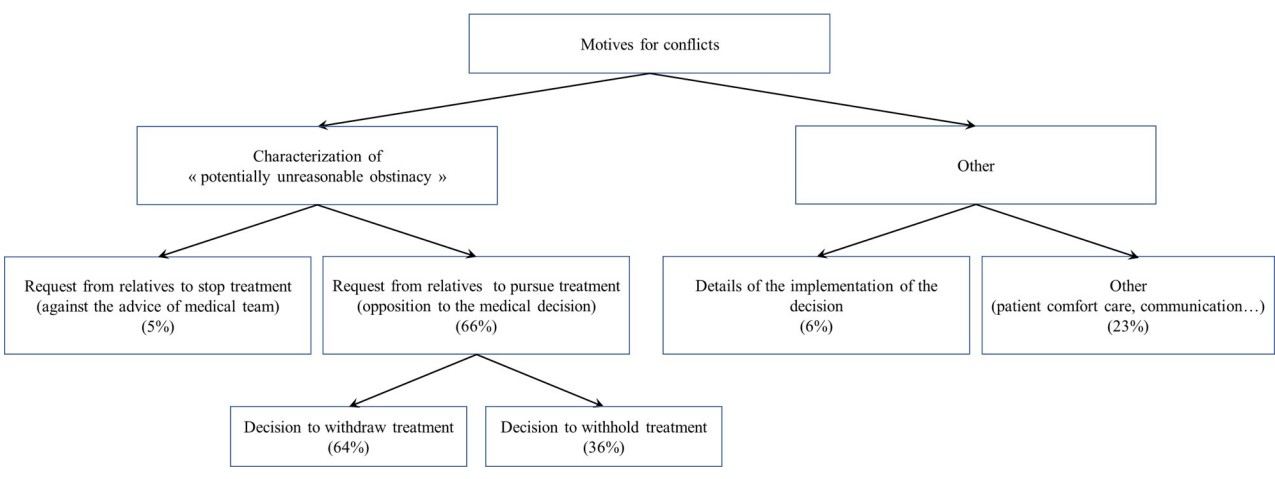

**Fig 1. Motives for the conflict.**

"information booklet" dedicated to LST limitation decision and comfort care. However, physicians did not support greater involvement of family members in patient care and medical decisions. Results are displayed in Fig 3.

## Secondary objectives

**Potential factors leading to conflict.** Whatever the motive for the conflict, several potential contributing factors were reported. Physicians' views on potential conflict-promoting factors are presented in S1 Fig. The absence of ADs, the lack of communication between caregivers and relatives, the multitude of relatives or the existence of intra-family disagreements, and the denial or misunderstanding of the medical situation were frequently mentioned as being implicated in creating the conflict. Religious, cultural or ethnic issues were also often mentioned.

The specific data related to the knowledge of the patient's wishes are presented in S2 File.

**Perceived consequences on caregivers and patient's care.** The answers concerning the consequences of the conflicts on the caregivers and on the patient's care are presented in S2 Fig. Physicians reported major consequences on the psychological exhaustion of caregivers, the meaning of their work and on medical practice.

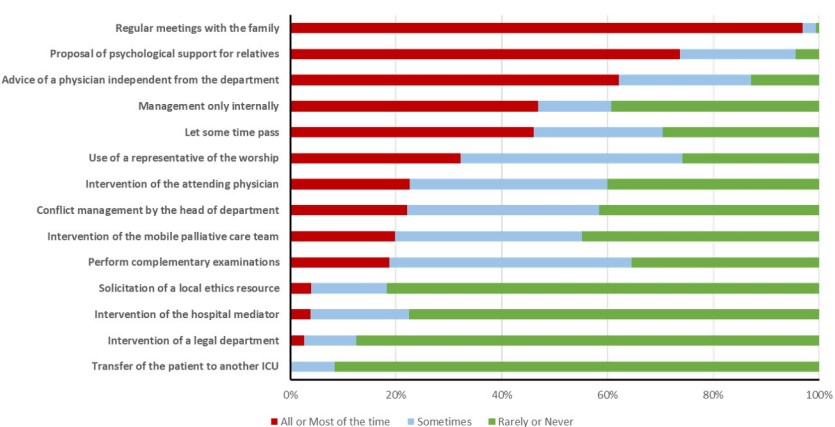

**Fig 2. Conflict management tools.** ICU, Intensive Care Unit. Data are expressed as percentage.

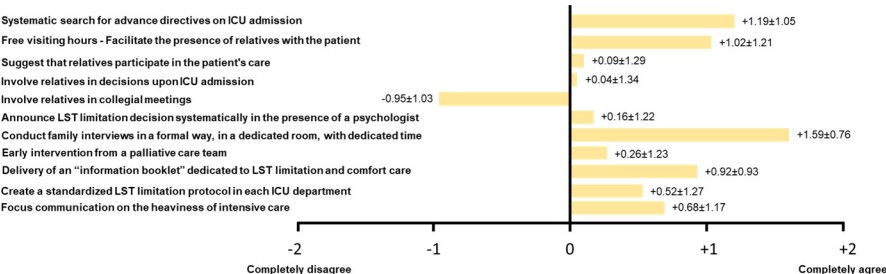

**Fig 3. Potential ways to prevent conflicts.** ICU, Intensive Care Unit; LST, Life Sustaining Therapies. Data are expressed as mean (±SD) on a Likert scale ranging from "completely disagree" (-2) to "completely agree" (+2).

**Impact of physician grade.** The answers to the questionnaire were analyzed according to the grade of the responding physicians (junior *vs* senior physicians). Younger physicians seem to face less conflict than more experienced physicians (p<0.01). However, junior physicians report being less likely to let time pass (12% *vs* 51%, p<0.01) and more frequently applying the decision without taking the conflict into account than senior physicians (47% *vs* 18%, p = 0.01). They also more often express a feeling of failure of medical care (81% *vs* 56%, p = 0.04).

## Discussion

This study highlights the issues that can be at the root of team-family conflicts during LST limitation decisions. The main results of this study underline that in the vast majority of cases, conflicts are related to requests from the relatives to continue treatments deemed unreasonable by the caregivers. Potential factors favoring these requests from relatives were identified, such as a lack of knowledge of the patient's wishes, religious or cultural issues, and a lack of communication. Furthermore, once the conflict is established, it could be relevant to improve its management, notably by making palliative care teams, ethics committees or mediation teams more accessible.

It is established that LST limitation decisions occur in more than half of the deaths that occur in intensive care [3, 23]. In the majority of cases, the relatives agree with LST limitation decisions but in rare cases, relatives can be opposed to the decision, opening the door to potential conflict. While physicians in our study report that these conflicts are infrequent, it was reported in the Conflicus Study that among the conflicts that arise between relatives and caregivers in ICU, EOL issues are one of the primary causes [21]. Other studies have also reported that situations in which LST limitation is being considered are frequently experienced as conflictual, both on the side of caregivers and relatives [24–26].

Our work highlights that conflicts arise from a disagreement between relatives and caregivers on the notion of "potential unreasonable obstinacy" when the former consider it appropriate to continue treatment while the latter do not. Several factors could explain this dissensus. Whereas the notion of unreasonable obstinacy is defined in French law as treatments "which appear to be useless, disproportionate or having no other effect than the artificial maintenance of life" [8], the criteria that define it are subjective and appeal to the values and beliefs of each individual [13]. In particular, individuals (patients, physicians, relatives) may have different views of quality of life, suffering and proportionality of treatments [27]. In this context, the patient is the best placed to determine the threshold beyond which treatments are "potentially inappropriate". By analogy with definitions of "futility" proposed in the past, treatments

become "unreasonable" when they are ineffective in achieving a goal, determined by the values and interests of the patient [28, 29].

However, a large proportion of patients admitted in ICU do not have formalized ADs and are unable to express themselves, in which case their own wishes cannot be known [3, 30, 31]. Our study shows that this lack of knowledge is one of the factors leading to potential conflict. In such cases, the only solution for the ICU physician is to refer to family members, which potentially opens the door to a conflict of values. For example, our results show that religious, cultural or ethnic issues could be associated with many conflict situations and other studies have shown that it appears to be an important dimension of EOL decisions [32, 33]. As this type of elements described in our study are potentially subjective [13], we can assume that a disagreement on the profound meaning of the hypothesis of withholding or withdrawing treatment may lead to the conflicts we describe.

Furthermore, our results suggest that prognostic uncertainty and hope for recovery (or a denial of the medical situation) could also be important factors in the team-family disagreement. Uncertainty is common in the practice of medicine, particularly with regard to EOL decisions [34]. In line with our results, a recent qualitative study exploring team-family conflicts in the ICU underlined the role of uncertainty about the patients' diagnosis or prognosis in reinforcing conflicts [35]. Indeed, families' misperceptions and misinterpretation of information can lead to differing expectations by physicians and family and often coincide with their disagreement with the proposed treatment decision [35–38]. These different expectations about prognosis seem to be common and an association with the beliefs of relatives, especially religious, has been described in the literature [37].

Once this disagreement of principles and values is established, our results suggest that the lack of communication potentiates it into a real conflict. The breakdown of communication between physicians and family-members was frequently mentioned in our results as favoring conflicts, probably insofar as it blocks any possibility for family-members and physicians to find a consensus on the notion of unreasonable obstinacy. Improving communication between physicians and relatives is essential in intensive care, especially in EOL situations [30, 39]. This has been shown to be key to improving the quality of the EOL decision-making process [40, 41]. The lack of communication could also be exacerbated by the lack of intervention of mobile palliative care teams, hospital mediator or local ethics resources observed here, even though this has been shown to be an effective solution to resolve conflict in the past [38].

EOL decisions are often difficult, and must involve the physicians, non-medical caregivers, the patient and the family-members to make the "right" decision [42]. When the patient has lost the capacity to judge quality of life or suffering, it debatable whether the physician has the moral authority to unilaterally make EOL decisions (27). One solution, which is very rarely used by the physicians interviewed in our study, is clinical ethics consultation. This is one of the key elements proposed in other countries to resolve conflicts [19]. In France, this practice is also garnering increasing interest, and its widespread generalization could be a valuable solution [43]. In case of conflict, clinical ethics consultation is a good way to involve all the different stakeholders (physicians, non-medical caregivers, patient, relatives, psychologists..) in making a decision where the notions of autonomy, beneficence/non-maleficence and justice are extensively discussed, in the full respect of the patient's values [44–46].

Team-family conflicts can have a major impact. For caregivers, our results suggest a major psychological impact with a risk of burnout, loss of motivation and increased work anxiety. Interestingly, the Conflicus Study [21] suggested an association between the fatigue felt at work by caregivers and the severity of conflicts. Other studies have also highlighted the risk of burnout or even resignation among caregivers confronted with conflicts, especially as the number, duration or severity of conflicts increases [47–49].

Finally, to limit the risk of conflict, the physicians we interviewed suggest a standardization of practices. On the one hand, this would involve optimizing the conditions for collecting the patient's wishes, in particular by systematically looking for the ADs (written or not). A more concerted effort to find out what the patient's wishes were, for example by striving to promote ADs, advance care planning and collegial decisional processes throughout care management, would be worth pursuing, to limit the risk of conflict [9, 50]. In this regard, the patient's health-care goals should probably be better anticipated and defined, especially for patients with chronic disease, by regularly discussing the patient's wishes in light of the therapeutic possibilities [51]. On the other hand, according to our results, improving the conditions of communication around EOL decisions seems essential and could help to reach agreement on the threshold of unreasonable obstinacy. In this regard, the need to improve clinicians' communication skills for eliciting and incorporating patients' values and preferences into treatment decisions has been underlined [31]. However, while physicians seem inclined in our study to encourage the presence of family members with patients, they do not seem to be in favor of involving the family members in care or in medical decisions, and they also do not think that it would reduce the risk of conflict. Thus, the question of how to include family members in medical decisions will continue to be of great interest in the future [30, 52].

Our study has some limitations. Firstly, this was a survey of physicians' reported practices and beliefs in France, and therefore, may not generalizable to other countries or cultures. However, the French experience is often cited as a model for reflection on the ethics and quality of end-of-life care in intensive care [53]. Secondly, we cannot exclude selection bias, in particular related to the dissemination of the study information through the RESC network. Indeed, participation was open and it is thus possible that only physicians with a particular interest in this issue answered the questionnaire. However, the variety of participants, as well as the diversity of the ICUs concerned are not in favor of a marked selection bias. Thirdly, nurses and nurses' aides were not invited to participate in this study. Non-medical caregivers are essential in LST limitation decisions and their opinions would have been interesting, as they are often witnesses to, or even involved in these conflicts. Nonetheless, the present study hypothesized that physicians are often on the front lines of conflict management and they were thus the first to be interviewed to meet the study objective. We intend to investigate the experiences of non-medical caregivers in a subsequent study. Fourth, only the physicians' opinions were solicited and this study does not assess the perception of conflict by the patients' families. Fifth, there was no precise definition of conflict given in this study. Definitions of conflict vary widely from one study to another [21, 54]. Insofar as our study did not focus on specific clinical situations, we felt that it was not beneficial to establish strong criteria for defining conflict, in order not to inadvertently orient the physicians' responses. Finally, our questionnaire included a limited number of questions in order to encourage participation in the study. Although the questionnaire was developed by a multidisciplinary team with different backgrounds, we cannot exclude the possibility that it was not exhaustive and that other unmeasured confounders were not taken into account.

## Conclusion

Requests to continue treatments deemed unreasonable by physicians are the main cause of team-family conflicts during LST limitation decisions. The implementation of the decision is most often suspended. Improved communication strategies and recommendations focused on the role of relatives in the decision-making process seem essential for the future.

## Supporting information

**S1 Table. English translation of the study questionnaire.**
(DOCX)

**S1 Fig. Physicians' views on factors leading to conflict.**
(TIF)

**S2 Fig. Impact of conflicts.** A. On caregivers; B. On patient care. Data are expressed as mean (±SD) on a Likert scale ranging from "completely disagree" (-2) to "completely agree" (+2).
(TIF)

**S1 File. Summary of the French LST limitation decision-making process.**
(DOCX)

**S2 File. Knowledge of the patient's wishes.**
(DOCX)

**S3 File. Minimal data set.**
(XLSX)

## Acknowledgments

We thank Delphine Pecqueur for her help in the data management.

## Author Contributions

**Conceptualization:** Mikhael Giabicani, Laure Arditty, Marie-France Mamzer, Isabelle Fournel, Jean-Philippe Rigaud, Jean-Pierre Quenot.

**Data curation:** Mikhael Giabicani, Laure Arditty.

**Formal analysis:** Mikhael Giabicani, Laure Arditty.

**Investigation:** Mikhael Giabicani, Laure Arditty.

**Methodology:** Mikhael Giabicani, Laure Arditty, Marie-France Mamzer, Nicolas Meunier-Beillard, Marta Spranzi, Jean-Philippe Rigaud, Jean-Pierre Quenot.

**Supervision:** Mikhael Giabicani, Jean-Philippe Rigaud, Jean-Pierre Quenot.

**Validation:** Marie-France Mamzer, Nicolas Meunier-Beillard, Emmanuel Weiss, Marta Spranzi, Jean-Philippe Rigaud, Jean-Pierre Quenot.

**Writing – original draft:** Mikhael Giabicani, Marie-France Mamzer, Isabelle Fournel, Fiona Ecarnot, Fabrice Bruneel, Emmanuel Weiss, Marta Spranzi, Jean-Philippe Rigaud, Jean-Pierre Quenot.

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
