## [Decision Letter · Decision Letter 0]

2 Feb 2023

PONE-D-23-00209Team-family conflicts over end-of-life decisions in ICU: a survey of practicesPLOS ONE

Dear Dr. Giabicani,

Thank you for submitting your manuscript to PLOS ONE. After careful consideration, we feel that it has merit but does not fully meet PLOS ONE’s publication criteria as it currently stands. Therefore, we invite you to submit a revised version of the manuscript that addresses the points raised during the review process.

We look forward to receiving your revised manuscript.

Kind regards,

Jean Baptiste Lascarrou

Academic Editor

PLOS ONE

Journal Requirements:

Additional Editor Comments:

In spite of an undeniable interest, the 3 reviewers have made remarks that need to be taken into account in full before to submit a new version of the manuscript.

Reviewers' comments:

Reviewer's Responses to Questions

**Comments to the Author**

1. Is the manuscript technically sound, and do the data support the conclusions?

Reviewer #1: No

Reviewer #2: Partly

Reviewer #3: Partly

2. Has the statistical analysis been performed appropriately and rigorously? 

Reviewer #1: I Don't Know

Reviewer #2: Yes

Reviewer #3: I Don't Know

3. Have the authors made all data underlying the findings in their manuscript fully available?

Reviewer #1: Yes

Reviewer #2: Yes

Reviewer #3: Yes

4. Is the manuscript presented in an intelligible fashion and written in standard English?

Reviewer #1: No

Reviewer #2: No

Reviewer #3: Yes

5. Review Comments to the Author

Reviewer #1: I thank Dr Giabicani Song for giving the opportunity to review the article entitled “Team-family conflicts over end-of-life decisions in ICU: a survey of practices”. The topic is important. However, I have major concerns about this research.

– The title is misleading. Indeed, the manuscript does not report “a survey of practices” (i.e. what is done), but “ a survey of French ICU physicians beliefs”.

– The methods of a questionnaire-based study on physician belief is not appropriate to study EOL practices in ICU and their consequences. A detailed description of decision making process should be provided. Non medical caregivers should be involved. A clear definition of conflict should be prospectively provided.

– The sample size calculation is not provided. The number of physcians included in the survey is quite small compared to the overall number of ICU physicians in France. How many ICUs were involved in the study? And how many physicians from each participating ICU were included in the survey?

– I disagree with authors stating “few data exist in the literature about situations of team-family conflicts regarding decisions to limit LST”. Many studies on conflicts in ICU and their cause have been reported for many decades. Please, see also this clinical review on the topic “Withdrawing life support and resolution of conflict with Families” published twenty years ago.

– The decision-making process is not reported. Involvement of relatives is poorly discussed. The reader is told only about what relatives “think” but not about the discussions, their number, who was involved… Involvement of non medical caregivers is a key issue in EOL decisions and is not discussed at all (excepted in the limitation section).

As a consequence, the manuscript adds nothing to the current knowledge to the topic. It is also hard for the reader to identify clear perspectives.

Last, the manuscript is too long and could be shortened. Some examples:

o in the introduction section, paragraphs from lines 65 to 95 provide only general well known information on EOL decisions and could be strongly shortened.

o Methods: “definition of conflict” should be removed, as authors clearly state conflicts were not a priori defined in the current study.

o Discussion: the section from line 316 to line 345 does not discuss the results of the study and could be shortened.

Reviewer #2: Thank you for asking me to review this paper. The research theme, conflicts between ICU teams and relatives, is particularly relevant. The questionnaire used provides many results. The high participation rate increases the power of these results.

However, I would recommand that the manuscript be completely and extensively revised to provide a clear and mening full message.

- The aim of the study is not clear and well defined (« to conduct a survey »), particularly in the abstract. The objective, with the 4 themes developed, is very broad.

- The methodological choice of questionnaire construction should be based on qualitative research references. The choice of questioning physicians who are members of a research network in ethics in intensive care is an important bias. It would have been interesting to have a cross evaluation of the perceptions of the paramedical staff.

- The results are very (too?) numerous.

o Table titles are not concise.

o Missing data is questionable.

o It is difficult to understand why the results begin with the paragraph about the patient's wishes. We would like to know the motives for the conflicts first. For the results regarding the reasons for the conflict, we would like to have a figure that illustrates the physicians' responses. It is not clear whether the responses to the questionnaire were exclusive or not. Finally, we do not have a very clear idea of the motives for the conflicts.

o The results of the figures are too numerous and difficult to read. I would recommend to invert the color code and the arrangement between "rarely" and "most of the time". We do not understand why there are A B C in the figures. It is not known whether a response rate cut-off is used by the authors to consider a response as relevant

o Sometimes the terms used in the results immediately suggest an interpretation (« interestingly » for example)

Furthermore, the manuscript is too long. It would deserve a complete English revision with particular attention to the terms chosen and the maintenance of these terms throughout the manuscript

Reviewer #3: The paper faces the topic of teams and family conflicts during LST limitation decisions in adult ICU patients.

The topic is mainly about ethics, as indicated at lines 318-321

However, the matter is not elaborated enough in the questionnaire nor investigated in the discussion.

Thus, in general I would suggest to:

analyze the literature on the matter of ethical issues in ICU and the role of ethics consultation.

Specifically, to elaborate the sentence (320-323):

Whereas the notion of unreasonable obstinacy is defined in French law as treatments … the concept remains vague and subjective and appeals to the values and beliefs of each individual. Only the patient himself is in a position to determine the threshold beyond which treatments are “potentially inappropriate”.

1) If the concept is too vague, it is very difficult to talk and discuss it. The topic is instead very complex and presents different criteria.

2) If only the patient can define the meaning, what would be the role of doctors, caregivers, psychologists and ethics consultants? How do they behave with the subject, without him being replaced?

6. PLOS authors have the option to publish the peer review history of their article (what does this mean?). If published, this will include your full peer review and any attached files.

Reviewer #1: No

Reviewer #2: No

Reviewer #3: No

---

## [Author Response · Author response to Decision Letter 0]

13 Mar 2023

To: Dr. Lascarrou, Editor, PLoS One

Dieppe, 9th March 2023

Re: Manuscript # Submission ID PONE-D-23-00209

Dear Doctor Lascarrou,

Thank you for your review of our article cited in reference, and for giving us the opportunity to submit a revised version of our manuscript for further consideration.

We are also most grateful to the editor and reviewers for their comments and constructive criticism, which have allowed us to strengthen our manuscript and clarify the key messages.

As requested, we provide our point-by-point responses to all comments and suggestions, together with marked-up and clean versions of the revised manuscript. The first two reviewers asked to shorten the manuscript while the third asked to add information. We tried to do our best to accommodate these requests.

Thank you in advance for your consideration of our revised manuscript, and we look forward to hearing back from you again in due course.

Yours sincerely,

Mikhael Giabicani, MD

 

Reviewer #1: I thank Dr Giabicani Song for giving the opportunity to review the article entitled “Team-family conflicts over end-of-life decisions in ICU: a survey of practices”. The topic is important. However, I have major concerns about this research.

We thank the reviewer for the useful comments and suggestions, which have enabled us to improve our manuscript and clarify the key messages. We have revised our manuscript according to your suggestions as far as possible.

Comments:

– The title is misleading. Indeed, the manuscript does not report “a survey of practices” (i.e. what is done), but “a survey of French ICU physicians beliefs”.

Thank you for pointing this out. In line with the Reviewer’s suggestion, we have modified the title to the following: 

“Team-family conflicts over end-of-life decisions in ICU: a survey of French physicians’ beliefs.”

– The methods of a questionnaire-based study on physician belief is not appropriate to study EOL practices in ICU and their consequences. 

We thank the reviewer for these comments. Insofar as we have re-focused the objective of the study on the source and management of the conflict, we believe that this part of the study can indeed be adequately addressed by a questionnaire survey.

A detailed description of decision making process should be provided. 

We have added a description of the French decision-making process in the supplementary files. This is now well codified in France by several successive pieces of legislation (laws dating from March 4, 2002; April 22, 2005; and February 2, 2016).

Non medical caregivers should be involved. 

We fully agree with the Reviewer that the involvement of non-medical caregivers would be important, and that their answers would have been very interesting. Nonetheless, in the present study, we hypothesized that physicians were often on the front line of conflict management with families in the ICU setting. Therefore, the physicians were the first to be interviewed on this issue. We intend to investigate the opinions and experiences of non-medical caregivers in a second study. We have emphasized this limitation in the discussion section.

A clear definition of conflict should be prospectively provided.

We acknowledge that we did not use a strict and precise definition of conflict. Indeed, the definition of “conflict” varies widely from one study to another [1,2]. Insofar as our study did not focus on specific clinical situations but aimed to report on medical practices in general terms, we believe that a precise definition might have unduly oriented the physicians’ responses, and therefore, would have introduced potential bias.

To take the Reviewer’s remark into account, we have removed the conflict definition paragraph and we have added this point to the discussion section.

– The sample size calculation is not provided. The number of physcians included in the survey is quite small compared to the overall number of ICU physicians in France. How many ICUs were involved in the study? And how many physicians from each participating ICU were included in the survey?

The Reviewer raises a pertinent point. We did not perform a priori sample size calculation, as this was not a randomized trial. However, we agree with the Reviewer that the number of physicians is small compared to the overall number of ICU physicians in France. Nevertheless, the number of responses obtained is comparable to that reported in previous work by our group, published in this journal, and using similar methodology [3,4].

In the present study, a total of 85 ICUs were involved, and on average, about 2 physicians per center responded. We have added this information in the results section. Despite the overall number of respondents being relatively low, the recruitment of centres was nonetheless representative of ICUs across France, with centres of varying specialty, size and work practice.

– I disagree with authors stating “few data exist in the literature about situations of team-family conflicts regarding decisions to limit LST”. Many studies on conflicts in ICU and their cause have been reported for many decades. Please, see also this clinical review on the topic “Withdrawing life support and resolution of conflict with Families” published twenty years ago.

We thank the reviewer for bringing this useful publication to our notice. To take the Reviewer’s comment into account, we have removed the cited sentence from the introduction, and emphasized the specificities of the French model, notably the absence of French recommendations on conflict management. We have also specified the purpose of the study and the data that warranted further exploration, in relation to the existing literature. 

– The decision-making process is not reported. Involvement of relatives is poorly discussed. The reader is told only about what relatives “think” but not about the discussions, their number, who was involved… Involvement of non medical caregivers is a key issue in EOL decisions and is not discussed at all (excepted in the limitation section).

We thank the reviewer for this comment. We recognize that this survey does not cover all the issues inherent in team-family conflicts. Firstly, the questionnaire was developed after an exploratory phase conducted as a qualitative study with a panel of ICU physicians, two consultants in clinical ethics and a professor of medical ethics. Secondly, we did not analyze specific clinical situations. Such individual analysis would have enabled us to report precise information on the involvement of relatives and non-medical caregivers in decision-making. These data will be explored in a specific study that we are currently setting up, and which will be initiated in the coming months. Thirdly, as mentioned above, we have re-focused the study’s objective and main results on what is relevant to physicians’ practices and placed the results related to physicians’ beliefs, such as what relatives “think”, in the supplemental data.

Finally, as mentioned in comment 2, we agree that the non-medical caregivers are essential and that their answers would have been very interesting. We have added to the discussion by underlining the involvement of relatives and non-medical caregivers in EOL decisions.

As a consequence, the manuscript adds nothing to the current knowledge to the topic. It is also hard for the reader to identify clear perspectives.

We thank the Reviewer for the pertinent comments and hope that the corrections will highlight clear perspectives.

Last, the manuscript is too long and could be shortened. Some examples:

o in the introduction section, paragraphs from lines 65 to 95 provide only general well known information on EOL decisions and could be strongly shortened.

o Methods: “definition of conflict” should be removed, as authors clearly state conflicts were not a priori defined in the current study.

o Discussion: the section from line 316 to line 345 does not discuss the results of the study and could be shortened.

Thank you for these useful suggestions, we have reduced the manuscript accordingly.

Reviewer #2: 

Thank you for asking me to review this paper. The research theme, conflicts between ICU teams and relatives, is particularly relevant. The questionnaire used provides many results. The high participation rate increases the power of these results.

We thank the Reviewer for the positive appreciation of our work and for the pertinent comments, which will help to strengthen our manuscript. Our point-by-point responses are given below. 

- The aim of the study is not clear and well defined (« to conduct a survey »), particularly in the abstract. The objective, with the 4 themes developed, is very broad.

We thank the reviewer for this comment. As suggested by the Reviewer and also by another Reviewer, we now emphasize the specificities of the French model in the introduction, notably the absence of recommendations on conflict management. We also give a clearer statement of the study aim, and the data that warranted further exploration, in relation to the existing literature.

We have rephrased the primary objective to focus on the object of the conflict, and on the issues that relate to physicians’ practices (conflict management). We have moved the content related to physicians’ beliefs to the secondary objectives.

We also clarified the aim of the study in the abstract.

- The methodological choice of questionnaire construction should be based on qualitative research references. 

The Reviewer raises a valid point, with which we fully agree. The development of the questionnaire should be based on qualitative research. We conducted an exploratory phase, which produced the empirical data on which the questionnaire items were based. This was conducted as a qualitative study using semi-directed interviews. This methodological point has been clarified in the manuscript. As suggested by the Reviewer, we also added suitable references.

- The choice of questioning physicians who are members of a research network in ethics in intensive care is an important bias.

We agree with the Reviewer that the distribution of the questionnaire within a research network in ethics in intensive care may introduce potential for bias. However, in the same way as research networks on sepsis or mechanical ventilation, the Research Network in Ethics in Critical Care (“RESC”) is more generally speaking a network for disseminating information and calling for participation in studies on the topic of ethics in critical care, than a network of physicians specializing in this field. In addition, the physicians in the research network were free to distribute the questionnaire to other physicians not referenced in the RESC. Also, dissemination through this network enabled us to achieve greater representativeness in terms of the type and size of participating ICUs, as well as in terms of ICU physician practices. It further ensured a higher response rate. We have emphasized this point in the Methods section and in the Discussion.

- It would have been interesting to have a cross evaluation of the perceptions of the paramedical staff.

Again, the Reviewer raises a pertinent point that was also raised by Reviewer #1. We fully agree that the perceptions of the paramedical staff are essential, and that their views would have been very interesting. This is an undeniable limitation of our survey. Nonetheless, we hypothesized that physicians were often on the front line of conflict management with families in the ICU setting, and they were thus the first to be interviewed on this issue. We intend to pursue our work, and interview non-medical caregivers in a second project that will comprise a qualitative survey (to develop the questionnaire) and then a quantitative study (using the questionnaire).

We have emphasized this limitation in the discussion.

- The results are very (too?) numerous.

o Table titles are not concise.

Thank you for pointing this out. We have simplified the titles, tables and figures and have moved some results to the supplementary files.

o Missing data is questionable.

The Reviewer raises a good point. However, it should be noted that the missing data only concerned the characteristics of the responding physicians. The results were complete for the questionnaire items about conflict management. Indeed, to encourage participation in the survey, we did not make the questions about their characteristics obligatory, in order to avoid discouraging the participation of potential respondents who did not want to reveal personal data. For the rest of the survey, there was no missing data. We have added these points in the Methods section.

o It is difficult to understand why the results begin with the paragraph about the patient's wishes. We would like to know the motives for the conflicts first. For the results regarding the reasons for the conflict, we would like to have a figure that illustrates the physicians' responses. It is not clear whether the responses to the questionnaire were exclusive or not. Finally, we do not have a very clear idea of the motives for the conflicts.

We thank the Reviewer for this pertinent suggestion. We have moved the motives for conflict to the beginning of the results section. The information about the patient's wishes has been moved to the supplementary data, since this was not part of the primary objective.

As suggested by the Reviewer, we added a new figure that illustrates the physicians’ responses regarding the reasons for the conflict. We clarified the motives for the conflicts in the manuscript.

Finally, the responses to the questionnaire were exclusive. There were no missing data (except for the characteristics of the ICUs and physicians). Physicians were also given the opportunity to insert free-text comments if they wished to clarify their response.

o The results of the figures are too numerous and difficult to read. I would recommend to invert the color code and the arrangement between "rarely" and "most of the time". We do not understand why there are A B C in the figures. It is not known whether a response rate cut-off is used by the authors to consider a response as relevant.

We thank the Reviewer for raising these pertinent comments about the figures. As also suggested by Reviewer #1, we have refocused the main objective of the study and moved some of the results to the supplementary files.

We have simplified figures to make them easier to read.

As suggested by the Reviewer, we have inverted the color code and the arrangement between “rarely” and “most of the time”.

We simplified Figure 1 by removing items that were redundant and by putting all the items on a single figure.

For responses on Likert scales, we considered a response rate to be relevant when it was above 50%. We have added this point in the Methods section.

o Sometimes the terms used in the results immediately suggest an interpretation (« interestingly » for example)

As suggested by the Reviewer, we have removed all terms of interpretation from the results section.

Furthermore, the manuscript is too long. It would deserve a complete English revision with particular attention to the terms chosen and the maintenance of these terms throughout the manuscript.

Again, the Reviewer raises a valid point that was also raised by Reviewer #1. We have shortened the manuscript to focus on clear perspectives. Finally, the manuscript has been thoroughly revised by a native English-speaking researcher.

Reviewer #3: 

The paper faces the topic of teams and family conflicts during LST limitation decisions in adult ICU patients. 

The topic is mainly about ethics, as indicated at lines 318-321

However, the matter is not elaborated enough in the questionnaire nor investigated in the discussion.

Thus, in general I would suggest to:

analyze the literature on the matter of ethical issues in ICU and the role of ethics consultation.

We thank the Reviewer for this useful suggestion. We have added a paragraph to the discussion about the undeniable value of clinical ethics consultation in cases of conflict.

Specifically, to elaborate the sentence (320-323):

Whereas the notion of unreasonable obstinacy is defined in French law as treatments … the concept remains vague and subjective and appeals to the values and beliefs of each individual. Only the patient himself is in a position to determine the threshold beyond which treatments are “potentially inappropriate”.

1) If the concept is too vague, it is very difficult to talk and discuss it. The topic is instead very complex and presents different criteria.

We thank the Reviewer for raising this point. We have clarified this point in the discussion about the determination of unreasonable obstinacy.

2) If only the patient can define the meaning, what would be the role of doctors, caregivers, psychologists and ethics consultants? How do they behave with the subject, without him being replaced?

Again, the Reviewer raises an important point. We have added a paragraph on clinical ethics consultations, and their utility in balancing the principles of autonomy, beneficence and justice from the perspective of each stakeholder (physicians, caregivers…), while focusing the reflection on the patient’s values.

References for the Responses to Reviewers: 

1. Azoulay E, Timsit JF, Sprung CL et al. (2009) Prevalence and factors of intensive care unit conflicts: the conflicus study. Am J Respir Crit Care Med 180:853-860. doi:10.1164/rccm.200810-1614OC

2. Way J, Back AL, Curtis JR (2002) Withdrawing life support and resolution of conflict with families. BMJ 325:1342-1345. doi:10.1136/bmj.325.7376.1342

3. Quenot JP, Jacquier M, Fournel I, Meunier-Beillard N, Grange C, Ecarnot F, Labruyere M, Rigaud JP, group RS (2023) Non-beneficial admission to the intensive care unit: A nationwide survey of practices. PLoS One 18:e0279939. doi:10.1371/journal.pone.0279939

4. Rigaud JP, Giabicani M, Meunier-Beillard N, Ecarnot F, Beuzelin M, Marchalot A, Dargent A, Quenot JP (2018) Non-readmission decisions in the intensive care unit under French rules: A nationwide survey of practices. PLoS One 13:e0205689. doi:10.1371/journal.pone.0205689

---

## [Decision Letter · Decision Letter 1]

10 Apr 2023

Team-family conflicts over end-of-life decisions in ICU: a survey of French physicians’ beliefs

PONE-D-23-00209R1

Dear Dr. Giabicani,

We’re pleased to inform you that your manuscript has been judged scientifically suitable for publication and will be formally accepted for publication once it meets all outstanding technical requirements.

Kind regards,

Jean Baptiste Lascarrou

Academic Editor

PLOS ONE
---

## [Editor Report · Acceptance letter]

14 Apr 2023

PONE-D-23-00209R1 

Team-family conflicts over end-of-life decisions in ICU: a survey of French physicians’ beliefs 

Dear Dr. Giabicani:

I'm pleased to inform you that your manuscript has been deemed suitable for publication in PLOS ONE. Congratulations! Your manuscript is now with our production department. 

Kind regards, 

on behalf of

Dr. Jean Baptiste Lascarrou 

Academic Editor

PLOS ONE